# *Plasmodium* infection disrupts the T follicular helper cell response to heterologous immunization

**Mary F Fontana[1]\*, Erica Ollmann Saphire[2], Marion Pepper[1]\***

[1]Department of Immunology, University of Washington School of Medicine, Seattle, United States; [2]Center for Infectious Disease and Vaccine Research, La Jolla Institute for Immunology, La Jolla, United States

**Abstract** Naturally acquired immunity to malaria develops only after many years and repeated exposures, raising the question of whether *Plasmodium* parasites, the etiological agents of malaria, suppress the ability of dendritic cells (DCs) to activate optimal T cell responses. We demonstrated recently that B cells, rather than DCs, are the principal activators of CD4[+] T cells in murine malaria. In the present study, we further investigated factors that might prevent DCs from priming *Plasmodium*-specific T helper cell responses. We found that DCs were significantly less efficient at taking up infected red blood cells (iRBCs) compared to soluble antigen, whereas B cells more readily bound iRBCs. To assess whether DCs retained the capacity to present soluble antigen during malaria, we measured responses to a heterologous protein immunization administered to naïve mice or mice infected with *P. chabaudi*. Antigen uptake, DC activation, and expansion of immunogen-specific T cells were intact in infected mice, indicating DCs remained functional. However, polarization of the immunogen-specific response was dramatically altered, with a near-complete loss of germinal center T follicular helper cells specific for the immunogen, accompanied by significant reductions in antigen-specific B cells and antibody. Our results indicate that DCs remain competent to activate T cells during *Plasmodium* infection, but that T cell polarization and humoral responses are severely disrupted. This study provides mechanistic insight into the development of both *Plasmodium*-specific and heterologous adaptive responses in hosts with malaria.

## Editor's evaluation

Malaria is still one of the world's most deadly diseases. Here, by using animal models of malaria infection and vaccination, authors have nicely showed that Dendritic cells (DCs) have a lower ability to uptake Plasmodium (the causative agent of malaria)-infected RBCs (particulate antigen), thereby being insufficient to activate T cells. This could be a potential reason for why our bodies cannot make appropriate acquired immunity upon Plasmodium infection.

## Introduction

Malaria, caused by parasites of the genus *Plasmodium*, is a leading global driver of infection-related mortality, resulting in 241 million cases of disease and over 600,000 deaths in 2020 (*World Malaria Report, 2021*). Clinical immunity to malaria (i.e. protection from symptoms) develops only after many years and repeated infections. Consequently, children in endemic areas remain vulnerable to severe illness and death for several years (*Black et al., 2010*; *Schofield and Mueller, 2006*; *Ryg-Cornejo et al., 2016b*), making the slow and incomplete acquisition of immunity a matter of grave concern for human health. The mechanisms underlying this apparently poor development of immunity have

\*For correspondence:
maryffontana@gmail.com (MFF);
mpepper@uw.edu (MP)

**Competing interest:** The authors declare that no competing interests exist.

been the subject of intense research for decades. Antigenic diversity among *Plasmodium* parasites is likely one factor; additionally, however, evidence suggests that immune responses generated in hosts with malaria may be defective or sub-optimal, relative to responses against immunizations or other pathogens (***Portugal et al., 2013***). For example, antibodies against *P. falciparum* in naturally exposed children are short-lived relative to antibodies against unrelated vaccine antigens (***Portugal et al., 2013***; ***Crompton et al., 2010***), and *P. falciparum* infection has been linked to diminished T helper cell responses in children and adults (***Ho et al., 1986***; ***Ho et al., 1988***; ***Chizzolini et al., 1990***; ***Rhee et al., 2001***; ***Bejon et al., 2009***). These observations raise the question of whether *Plasmodium* parasites subvert optimal immune responses to facilitate repeated infection of their mammalian hosts. The immune defects associated with *Plasmodium* infection are not limited to responses against the parasite itself: children with acute malaria exhibit impaired responses to a range of vaccinations (***Greenwood et al., 1972***; ***Williamson and Greenwood, 1978***), and *Plasmodium* infection causes declines in pre-existing, unrelated vaccine-induced antibodies and plasma cells in both mice and humans (***Banga et al., 2015***; ***Ng et al., 2014***). From a public health perspective, it is therefore also important to understand how malaria negatively impacts heterologous immunizations.

One proposed mechanism for the development of sub-optimal or short-lived immune responses in malaria is that *Plasmodium* parasites may suppress the activity of dendritic cells (DCs), innate immune cells that initiate canonical T cell responses to microbes (evidence for malaria-induced dysfunction reviewed in ***Wykes and Good, 2008***). Some studies have found that *P. falciparum*-infected RBCs (iRBCs) or parasite products can render human DCs refractory to activation and restrict priming of CD4[+] T cells in vitro (***Urban et al., 1999***; ***Pouniotis et al., 2004***; ***Elliott et al., 2007***; ***Yap et al., 2019***; ***Millington et al., 2006***; ***Pack et al., 2021***), and humans with malaria have fewer and less activated circulating DCs than healthy controls (***Pinzon-Charry et al., 2013***; ***Woodberry et al., 2012***). However, these decreases in circulating DCs during infection could reflect recruitment to infected tissues, and additional in vitro studies have concluded that DCs *do* become activated and can prime T cells after exposure to iRBCs and parasite products (***Coban et al., 2005***; ***Griffith et al., 2009***; ***Götz et al., 2017***). Overall, the question of whether malaria suppresses DC activity remains a subject for debate (***Wykes and Good, 2008***; ***Struik and Riley, 2004***).

Mouse models have provided additional characterizations of DC function during *Plasmodium* infection, but with similarly conflicting results. Some studies have reported that *Plasmodium* suppresses DC function under specific circumstances (***Wykes et al., 2007a***; ***Sponaas et al., 2012***; ***Wykes et al., 2007b***), but others have established that DCs isolated from infected mice can present parasite-derived antigens to activate T cells and generate protective responses (***Pouniotis et al., 2004***; ***Wykes et al., 2007a***; ***Sponaas et al., 2012***; ***Sponaas et al., 2006***; ***Voisine et al., 2010***; ***Perry et al., 2004***). Whether they actually perform this function in hosts with malaria is less clear. Several papers have sought to test a role for DCs in vivo using a transgenic mouse model that permits deletion of CD11c-expressing cells. These studies concluded that conventional CD11c[hi] DCs were important for antigen presentation and T cell responses in malaria (***Borges da Silva et al., 2015***; ***Ueffing et al., 2017***). However, CD11c is also upregulated on activated B cells, making the results of these studies difficult to interpret. Using more selective genetic tools to ablate antigen presentation specifically in DCs or B cells, we recently demonstrated that B cells, rather than DCs, are the dominant antigen-presenting cells (APCs) that direct activation and polarization of the CD4[+] T cell response in mice with malaria (***Arroyo and Pepper, 2020***). In light of this finding, we considered it important to revisit the question of DC dysfunction in mice with malaria.

Historically, the ability of B cells to present antigen to T cells has been studied primarily in the context of cognate interactions with CD4[+] cells that have already been activated by DCs. However, B cells can also interact with and activate naïve T cells directly. Hong et al. showed recently that whereas DCs were required for activating CD4[+] T cell responses to a soluble protein antigen, CD4[+] T cell responses to the same antigen delivered on a nanoparticle required presentation by B cells (***Hong et al., 2018***). In light of this finding, here we examined whether a differential ability to take up particle-like iRBCs might bias presentation of *Plasmodium* parasite antigens away from DCs and towards B cells. Further, we tested whether DCs in *Plasmodium*-infected mice remained competent to activate T cell responses to a heterologous, soluble protein antigen—an approach that allowed us to assess the functional capacity of DCs in hosts with malaria, independent of whether DCs can efficiently obtain iRBC-associated antigen in vivo. We found that DCs are relatively poor phagocytes of iRBCs, which

may be one contributing factor in the dominance of B cells as APCs in this disease. We also found that DCs remained competent to activate CD4+ T cell responses to soluble antigens in hosts with malaria, suggesting no global dysfunction occurs. However, CD4+ T cell polarization, germinal center formation, and antibody production were profoundly altered, a finding that has implications for vaccination schemes in malaria-endemic regions.

## Results

### DCs do not efficiently capture infected RBCs in vivo

We recently demonstrated that mice infected with the rodent-adapted parasite *P. yoelii* developed an antigen-specific CD4+ T cell response that was strongly polarized to a T follicular helper cell (Tfh) phenotype, specialized for assisting B cells with affinity maturation within germinal centers. In mice with malaria, these antigen-specific T cells possessed diminished proliferative capacity and longevity compared to T helper cells specific for the same epitope delivered in the context of an LCMV infection. Using genetically modified mice to selectively disrupt or restore antigen presentation in either DCs or B cells, we also determined that B cells, rather than DCs, were the primary APCs in this infection and orchestrated the striking Tfh polarization phenotype (*Arroyo and Pepper, 2020*).

We wished to better understand factors that favor B cells as the dominant APCs in this infection model, especially in light of recent work demonstrating that different APCs are required for soluble versus nanoparticle-associated antigen (*Hong et al., 2018*). Since *Plasmodium* infects RBCs, its antigens are primarily associated with 'particles' in vivo (i.e. RBCs). We hypothesized that the particulate nature of the parasite antigen itself might bias the APC response toward B cells and away from DCs, perhaps contributing to DCs' apparent lack of participation in antigen presentation to T cells in this infection (*Arroyo and Pepper, 2020*). Upon RBC invasion, parasite antigens can also be shed from the parasite surface into the circulation, where APCs might encounter them in soluble form (*Beeson et al., 2016*); however, the epitope we examined in our previous study was fused to a parasite protein expressed in the parasitophorous vacuole membrane (Hep17; *Charoenvit et al., 1999*), and therefore we expect APCs to interact with it primarily in the context of an iRBC. Another previous study found that DCs do interact with iRBCs in the spleens of infected mice (*Borges da Silva et al., 2015*), and indeed, it is well-established that DCs from infected mice *can* activate *Plasmodium*-specific T cell responses ex vivo, indicating that they must take up some parasite antigen (*Pouniotis et al., 2004*; *Sponaas et al., 2012*; *Sponaas et al., 2006*; *Voisine et al., 2010*; *Perry et al., 2004*). Yet the fact that they are capable of T cell activation ex vivo does not necessarily mean that they are the preferred or dominant APCs activating T cells in vivo, as infection could impact DC localization or migration. Some of these studies also examined B cells ex vivo and concluded that they did not activate CD4+ T cells during malaria; however, they specifically isolated the CD11c- subset, likely excluding the activated B cells that are most likely to serve as APCs (*Voisine et al., 2010*; *Perry et al., 2004*). Other studies either did not specifically examine B cells or deliberately excluded them from analysis. Given our recent findings on the central role of B cells in activating CD4+ T cell responses in this system, we considered it critical to revisit previous work examining uptake of iRBCs in the spleens of infected mice.

Modifying the experimental approach of *Borges da Silva et al., 2015*, we used flow cytometry to examine how well various splenic APC subsets, including B cells, could bind to labeled RBCs infected with the parasite *P. chabaudi*. Like *P. falciparum*, *P. chabaudi* exports membrane proteins that render iRBCs cytoadherent (*Stephens et al., 2012*), a characteristic found to be important for interaction with (and suppression of) DCs in one study (*Urban et al., 1999*). We enriched iRBCs from the blood of infected mice (*Figure 1—figure supplement 1A*), labeled them with a fluorescent dye, and injected them intravenously into naïve mice. Differentially labeled naïve RBCs (nRBCs) were injected for comparison. We harvested spleens after 30 min and assessed association of labeled nRBCs and iRBCs with various splenic subsets, including B cells (defined as CD45+ Thy1.2- MHCII+ B220+ CD11clo/int), DCs (CD45+ Thy1.2- MHCIIhi CD11chi), and red pulp macrophages (RPMs; CD45+ Thy1.2- MHCII+ F4/80hi), which specialize in uptake and recycling of senescent RBCs (*Figure 1A*). We found that DCs made up only a small fraction of total RBC+ splenocytes; instead, RBCs were mostly distributed between RPMs and B cells (*Figure 1B–D*). Additional flow experiments examining Siglec-H+ plasmacytoid DCs did not detect significant association of iRBCs with these cells (data not shown). The frequencies of iRBC+



**Figure 1.** DCs capture only a small fraction of infected RBCs in vivo. Red blood cells infected with *Plasmodium chabaudi* (iRBCs) were enriched from infected mice, labeled with a fluorescent dye, and injected intravenously into naïve C57Bl/6 J mice. Labeled naïve RBCs (nRBCs) were injected as a control. Splenocytes were analyzed 30 min later for fluorescent iRBC label by flow cytometry. (**A**) Representative gating of bulk dendritic cells (DCs), B cells, and red pulp macrophages (RPMs). (**B**) Representative plots showing the distribution of labeled iRBCs in DCs, B cells, and RPMs. (**C**) Quantification showing the percentage of the total RBC+ population that consisted of DCs, RPMs, or B cells as indicated. (**D**) Quantification of RBC+ DCs, RPMs, and B cells expressed as a percentage of total splenocytes. **A** and **B** depict representative plots from one of four independent experiments. **C** and **D** show pooled data (mean +/- SD) from all four experiments, with each symbol representing one mouse (n=12 mice for iRBC treatment, 14 for nRBC treatment). **, p<0.01 and ****, p<0.0001 by one-way ANOVA with Tukey's post-test. n.s., not significant.

The online version of this article includes the following source data and figure supplement(s) for figure 1:

**Source data 1.**

**Figure supplement 1.** Uptake of enriched iRBCs by B cells.

DCs that we measured were similar to those reported by *Borges da Silva et al., 2015*; however, that group did not perform a comparison with B cells.

In our in vivo experiments, DCs bound similar frequencies of nRBCs and iRBCs (*Figure 1D*). Interestingly, B cells bound significantly more iRBCs than nRBCs both in vivo and in vitro (*Figure 1C–D* and *Figure 1—figure supplement 1B*), although some apparent RBC binding to B cells in vivo was artifactual and occurred during processing (*Figure 1—figure supplement 1C, D*). To test whether B cell binding of iRBCs required the B cell receptor (BCR), we incubated nRBCs or iRBCs with transgenic B cells from MD4 mice, which express a BCR specific for hen egg lysozyme (*Goodnow et al., 1988*). This experiment revealed that preferential binding of iRBCs was independent of the BCR (*Figure 1—figure supplement 1B*), leading us to speculate that B cells may selectively interact with iRBCs through other, antigen-independent pathways such as complement receptors, Fc receptors, or scavenger receptors

(**Vijay et al., 2021**; **Shen et al., 2018**). Altogether, these experiments show that of all the iRBCs interacting with APCs in the spleen, only a small percentage are associated with DCs.

## DCs take up soluble antigen more efficiently than infected RBCs

Because DCs are well-established as the primary APCs that activate naïve T cells in response to soluble antigens, we next tested how DC uptake of iRBCs compared to uptake of soluble protein. We simultaneously injected the easily traceable fluorescent molecule phycoerythrin (PE) along with labeled iRBCs intravenously into mice and measured uptake by splenic APC subsets after 30 min. Very few splenocytes acquired both PE and iRBC labels in our experiments; this could be due both

**Figure 2.** DCs take up soluble antigen more efficiently than infected RBCs. Mice were intravenously injected with enriched, fluorescently labeled iRBCs together with PE and splenocytes were analyzed 30 min later for acquisition of fluorescent signal. (**A**) Representative gating of PE$^+$ or iRBC$^+$ splenocytes, further subsetted into the indicated APC populations. (**B**) Quantification of the distribution of PE and iRBC labeling among DCs, B cells, and RPMs. (**C**) Alternative gating strategy showing DCs, B cells, and RPMs in untreated mice (top row) and 30 min after injection of PE and iRBCs (bottom panel). (**D**) Quantification showing the percentage of each APC subset that acquired iRBC or PE label, as gated in (**C**). (**E**) Ratio of the percentage of each subset that acquired PE signal to the percentage of that subset that acquired iRBC signal. (**B, D, and E**) show pooled data (mean +/- SD) from two independent experiments (n=5 per group). **, p<0.01 and ***, p<0.001 by Mann-Whitney (**D**) or one-way ANOVA with Tukey's post-test (**E**). n.s., not significant.

The online version of this article includes the following source data for figure 2:

**Source data 1.**

to differences in a particular cell's ability to take up either label, and to the overall low frequency of splenocytes that pick up any label (making a double positive cell even more rare). Gating separately on all MHCII+ cells that carried either the PE or the iRBC label, we further divided the labeled cells into APC populations (*Figure 2A*). A majority of both PE+ and iRBC+ events were B cells, reflecting the abundance of this subset, which comprises ~50% of splenocytes. However, DCs accounted for around 10% of PE+ events, despite comprising only 1–2% of splenocytes. In contrast, the iRBC+ population included very few DCs (1–3%, commensurate with their total splenic frequency), but many more RPMs (~20%) in addition to B cells (*Figure 2A and B*).

As a second way of analyzing these same data, we examined the fraction of each APC subset that acquired either PE or iRBC label. Whereas ~5% of CD11c^hi MHCII^hi splenic DCs had taken up IV-injected PE, the percentage of DCs labeling with iRBCs was virtually zero (*Figure 2C and D*). We considered that this difference in uptake could be due to different effective doses of PE and iRBCs. However, DCs capture a greater fraction of the total cell-associated PE than the total cell-associated iRBC signal, indicating a better ability to capture PE (*Figure 2B*). Further, the ratio of PE+ DCs to iRBC+ DCs within each mouse was significantly higher than the ratio of PE+ RPMs or B cells to iRBC+ RPMs or B cells, respectively, indicating that DCs are biased toward taking up PE more so than iRBCs, relative to the other APC subsets examined (*Figure 2E*). Thus, DCs do not capture iRBCs as efficiently as soluble antigen in vivo. Instead, the majority of iRBCs associate with other APC subsets, including B cells.

It is possible that even a small amount of antigen uptake and presentation is sufficient to permit activation of naïve T cells, and indeed it has been shown that DCs from infected mice do acquire *Plasmodium* antigens in vivo (*Pouniotis et al., 2004*; *Sponaas et al., 2012*; *Sponaas et al., 2006*; *Voisine et al., 2010*; *Perry et al., 2004*). However, in light of our previous study identifying B cells as the primary APCs in mice with malaria (*Arroyo and Pepper, 2020*), as well as data showing that B cells preferentially present nanoparticle-associated antigens to T cells (*Hong et al., 2018*), we suggest that poor acquisition of iRBC-borne antigen may be one factor underlying the lack of substantial contribution from DCs to the antigen-specific CD4+ T cell response in *Plasmodium*-infected mice. It remains to be seen whether parasite antigens that are shed into the circulation might induce DC-dependent responses better than iRBC-associated epitopes such as the one we examined in our previous study (*Arroyo and Pepper, 2020*). Stephens et al. have reported that transgenic CD4+ T cells specific for the shed portion of the parasite coat protein MSP1 are capable of assisting B cells in antibody production (*Stephens et al., 2005*), but further characterization is needed to determine how these T cells are activated and polarized.

## Monocyte-derived DCs take up majority of soluble antigen in *Plasmodium*-infected mice

Our initial experiments raised the possibility that differential uptake of parasite antigen might contribute to the prominence of B cells in antigen presentation during malaria. However, they did not address the question of whether DC activation, antigen presentation, upregulation of costimulatory molecules, and/or secretion of cytokines and chemokines are suppressed during *Plasmodium* infection, as has been suggested (*Wykes and Good, 2008*; *Urban et al., 1999*; *Götz et al., 2017*; *Sponaas et al., 2012*). To probe the question of whether DCs retain the capacity to present antigen and activate T cells in mice with malaria, independent of their ability to access or take up particulate antigen, we investigated antigen uptake, DC activation state, and initiation of T cell responses to a heterologous, soluble protein antigen delivered to naïve or *P. chabaudi*-infected mice.

First, we tested whether DC uptake of a soluble protein was altered in infected mice. We injected PE into naïve mice or mice infected for 5 days with *P. chabaudi*. At this timepoint, *Plasmodium*-specific T cell differentiation is under way (*Arroyo and Pepper, 2020*) but germinal centers, which are delayed in malaria relative to their kinetics following protein immunization, have not yet formed (*Krishnamurty et al., 2016*). As expected, we found that parasitemia was still ascending (*Figure 3A*) but total DC numbers in the spleen were not significantly different from naïve mice (*Figure 3B*). 30 min after injection, we observed significantly higher frequencies of PE+ DCs in infected compared to uninfected mice (*Figure 3C*), indicating that there is no defect in acquisition of soluble antigen by DCs during infection. B cell uptake of PE was slightly increased in infected versus uninfected mice, but the difference was not statistically significant (*Figure 3D*). We next examined whether infection altered the phenotype or identity of PE+ DCs, since DC populations change during infection. Specifically, we

**Figure 3.** Monocyte-derived DCs take up majority of soluble antigen in *Plasmodium*-infected mice. (**A**) Mice were infected with $10^6$ *P. chabaudi*-parasitized RBCs and parasitemia was monitored by thin blood smear. (**B**) DCs were enumerated by flow cytometry in the spleens of mice at homeostasis or 5 days after *P. chabaudi* infection (Pc D5). (**C**) Naïve or day 5-infected mice were injected intraperitoneally (i.p.) with PE, and PE+ splenic DCs were quantified after 30 min by flow cytometry. Note log scale. (**D**) Mice were treated as in C, and PE+ splenic B cells were quantified after 30 minutes. (**E**) Representative gating of conventional CD64- DCs (cDC) and CD11b^hi CD64+ monocyte-derived DCs (moDC) in naïve or infected mice injected with PE. Top plots show all DCs; bottom plots show PE+ DCs. (**F**) Plots depicting the percentage of total or PE+ DCs that are cDC or moDC in naïve or infected mice. (**G**) Histograms and quantification showing expression (median fluorescent intensity, MFI) of MHCII, CD80, and CD86 on cDC and moDC. moDC are shown only from infected mice due to their scarcity in naïve mice. A shows data from one experiment (n=3), representative of many. B-D, F, and G show mean +/- SD from at least three experiments (n=6 per group for C, D; 10 uninfected, 8 infected for all others). *, p<0.05, **, p<0.01, ***, p<0.001, ****, p<0.0001 by Mann-Whitney (**B–D**) or one-way ANOVA with Tukey's post-test (**G**). n.s., not significant.

*Figure 3 continued on next page*

*Figure 3 continued*

The online version of this article includes the following source data for figure 3:

**Source data 1.**

measured the frequency of newly recruited monocyte-derived DCs (moDCs), which were defined as MHCII^hi CD11c^hi CD11b^hi CD64^+ (***Figure 3E***). moDCs were rare in the spleens of uninfected mice. In infected mice, they comprised ~30% of all DCs, but made up ~80% of PE^+ DCs (***Figure 3E and F***). Thus, moDCs disproportionately capture the majority of soluble antigen in infected mice, compared to CD64^- conventional DCs (cDCs).

Given this observation, we next questioned whether moDCs possessed the components to activate T cells. Flow cytometric analysis of surface markers revealed that moDCs expressed at least as much MHCII as cDCs, and had significantly higher expression of the costimulatory molecules CD80 and CD86 compared to cDCs from either uninfected or infected mice (***Figure 3G***). Altogether, our data demonstrate that DC uptake of soluble antigen remains intact after 5 days of *P. chabaudi* infection, but that the antigen-capturing subset shifts primarily from cDCs to moDCs. These moDCs express robust levels of MHCII and costimulatory molecules, and therefore possess the basic requirements for efficient T cell priming.

## Intact CD4^+ T cell expansion, but disrupted polarization, following heterologous immunization in infected mice

We next examined whether the DCs that took up soluble antigens in infected mice retained the capacity to activate CD4^+ T cells in vivo. For these experiments, we used a recombinant form of the glycoprotein (GP) from lymphocytic choriomeningitis virus (LCMV) because it contains the well-defined CD4^+ T cell epitope GP66, permitting CD4^+ T cells specific for this epitope to be detected with a fluorescently labeled peptide-MHCII tetramer (***Zander et al., 2017***). We previously characterized GP66-specific CD4^+ T cell responses in mice infected with a transgenic *P. yoelii* parasite that expresses this epitope (***Arroyo and Pepper, 2020***); thus, we have extensive data on the phenotype and kinetics of GP66-specific CD4^+ T cells when the epitope is delivered in the context of an iRBC. In the present study, administration of soluble recombinant GP with adjuvant during an infection with (non-transgenic) *P. chabaudi* allowed us to examine the same population of GP66-specific T cells as they responded to a soluble antigen presented by DCs, all within the context of a malaria-inflamed spleen.

We injected uninfected or day 5-infected mice with GP plus adjuvant and analyzed splenic GP66-specific CD4^+ T cells by flow cytometry 8 days after immunization (which was 13 days post-infection). GP66-specific T cells were rare in unimmunized mice but expanded dramatically following immunization in both uninfected and infected mice. No T cell expansion was observed in infected mice that did not receive GP. The total number of GP66-specific CD4^+ T cells was not significantly different between uninfected and infected mice following immunization of both, indicating that the ability of DCs to present soluble antigen and activate T cells was roughly intact in infected mice (***Figure 4A and B***).

We next examined the phenotype of GP66-specific T cells in uninfected and infected mice. Specifically, we measured polarization of CD4^+ T cells into CXCR5^+ PD-1^hi germinal center (GC) Tfh cells. As expected, uninfected mice immunized with GP developed a distinct population of GC Tfh cells, as well as a sizeable subset of CXCR5^+ PD-1^lo Tfh cells. Strikingly, however, the Tfh and GC Tfh populations were virtually absent among GP66-specific cells in infected mice following GP immunization (***Figure 4C and D***). This was not due to a generalized loss of Tfh cells in infected mice, since abundant CXCR5^+ T cells were observed in the bulk (non-tetramer-binding) T cell population (***Figure 4—figure supplement 1***) and our previous study found strong Tfh polarization of GP66-specific CD4^+ T cells at similar timepoints after infection with GP66-expressing parasites (***Arroyo and Pepper, 2020***). We examined whether this disruption persisted over time by quantifying antigen-specific T cells 3–5 weeks post-immunization. At these late timepoints, the total number of GP66-specific CD4^+ T cells was still not significantly different between mice that had concurrent *P. chabaudi* infection and those that did not, although there was a nonsignificant trend toward higher T cell numbers in infected mice (***Figure 4E***). The percentage of GP66-specific T cells exhibiting a GC Tfh phenotype remained sharply decreased in infected mice (***Figure 4F***), but the absolute number of GP66-specific GC Tfh cells was not significantly



**Figure 4.** CD4⁺ T cell expansion, but disrupted polarization, following heterologous immunization in infected mice. (**A–G**) Mice were immunized i.p. with recombinant LCMV glycoprotein (GP) plus adjuvant, either alone (GP Immunized) or 5 days after *P. chabaudi* infection (Infected +GP Immunized). GP66-specific CD4⁺ T cells were analyzed in spleen 8 days (**A–D**) or 23–35 days (**E–G**) after immunization. (**A**) Representative gating and (**B**) quantification of GP66⁺ CD4⁺ T cells from uninfected and infected mice with or without GP immunization. Note log scale in B. (**C**) Gating and (**D**) quantification of activated Tfh (CD44ʰⁱ CXCR5⁺) and GC Tfh (CD44ʰⁱ CXCR5⁺ PD-1ʰⁱ) GP66-specific cells. (**E–G**) Total GP66-specific CD4⁺ T cells (**E**) and the frequency (**F**) and absolute number (**G**) of GP66-specific GC Tfh cells were quantified 23–35 days post-immunization. (**H–J**) Mice left uninfected or infected for 5 days with *P. chabaudi* were immunized with irradiated *P. yoelii*-parasitized RBCs expressing GP66 (IR-PyGP66), and splenic T cells were analyzed 8 days later. (**H**) Quantification of total GP66-specific CD4⁺ T cells. (**I**) Representative flow plot of CXCR5 and PD-1 expression on GP66-specific T cells. (**J**) Frequencies of GP66-specific Tfh cells (left graph) and GC Tfh cells (right graph). A-D represent data from four independent experiments (pooled n=10 "GP" mice, 12 "Infected +GP" mice). **E–G** show pooled results from one Day 23 experiment and one Day 35 experiment (n=5 "GP" mice, 7 "Infected +GP" mice). **H–J** show pooled results from two independent experiments (n=5 immunized, 7 infected +immunized). Red numbers within flow plot gates represent the frequency of cells within the gate. *, p<0.05, **, p<0.01 and ****, p<0.0001 by one-way ANOVA with Tukey's post-test (**B**) or Mann-Whitney (others). n.s., not significant.

*Figure 4 continued on next page*

*Figure 4 continued*

The online version of this article includes the following source data and figure supplement(s) for figure 4:

**Source data 1.**

**Figure supplement 1.** Abundant CXCR5⁺ expression by bulk CD4⁺ T cells in infected mice.

different between the two experimental groups, perhaps because of the slightly higher numbers of total GP66-specific T cells in the infected mice (*Figure 4G*). Together, these data demonstrate that an established *P. chabaudi* infection does not prevent expansion of T cells in response to a heterologous soluble antigen, but does temporarily disrupt the polarization of antigen-specific T cells into B cell-helping Tfh and GC Tfh cells.

Previous work has shown that DCs can orchestrate initial differentiation of CXCR5⁺ Tfh cells 2–4 days after immunization, but that maintenance of the Tfh phenotype at day 8 requires both cognate and non-cognate interactions with B cells (*Kerfoot et al., 2011*; *Pepper et al., 2011*; *von der Weid and Langhorne, 1993*). Therefore, we hypothesized that interactions between DCs and T cells were intact in infected mice, allowing for T cell expansion, but that infection disrupted interactions between GP-specific T cells and B cells, preventing establishment of Tfh and GC Tfh populations. To further dissect B-T cognate interactions in the infected spleen, we treated uninfected mice, or mice infected for 5 days with *P. chabaudi*, with irradiated iRBCs containing nonreplicating transgenic GP66⁺ *P. yoelii* parasites (*Hahn et al., 2018*) and measured GP66-specific T cell responses 8 days later. Since iRBC-associated GP66 is primarily presented by B cells (*Arroyo and Pepper, 2020*), we reasoned that if interactions between newly activated B and T cells were disrupted in day 5-infected mice, expansion of GP66-specific CD4⁺ T cells should be defective. This was indeed what we observed (*Figure 4H*). CXCR5⁺ Tfh and GC Tfh subsets were also significantly diminished in infected mice, consistent with the loss of cognate and non-cognate B-T interactions that normally induce Tfh polarization (*Figure 4I and J*). Taken together, our data suggest that in an infected spleen, DCs remain competent for uptake and presentation of soluble antigens, but that the signals required for T cell polarization into Tfh cells—including both cognate and non-cognate interactions with B cells—are disrupted.

## B cell expansion and GC formation in response to heterologous immunization are curtailed in infected mice

Because antigen-specific GC Tfh cell responses are sharply diminished when antigen is administered during an established *P. chabaudi* infection (*Figure 4C and D*), we examined expansion of antigen-specific B cells and development of the GC B cell population after the same immunization scheme. A fluorescently labeled GP protein tetramer was used to identify GP-specific B cells (*Figure 5A*; *Kim et al., 2019*). These cells expanded robustly in number 8 days after GP immunization of uninfected mice, but not following GP immunization of mice infected for 5 days with *P. chabaudi* (*Figure 5A and B*). The frequency of CD138⁺ GP-specific plasmablasts was modest 8 days post-immunization and was not different between infected and uninfected mice (*Figure 5C*). In contrast, a sizeable population of GP-specific GC B cells (CD138⁻ CD38⁻ GL7⁺) appeared in uninfected, immunized mice, but was greatly decreased in infected mice following immunization (*Figure 5D*). Previous work from our group found that *Plasmodium*-specific GC B cells are readily detected at this time (i.e. 13 days after infection) (*Krishnamurty et al., 2016*). These data identify a functional consequence of the drastic decrease in GP66-specific GC Tfh cells in infected mice, as Tfh cells are required to sustain the repeated rounds of proliferation and selection that antigen-specific B cells undergo in GCs. Consistent with the decreases in total and GC B cell numbers, GP-specific serum IgG antibody titers were much lower in infected compared to uninfected mice 7 days after immunization (*Figure 5E and F*). A slight but significant defect in serum IgG persisted 14 days after immunization; however, after 3 weeks, titers in infected mice increased to the levels of those in uninfected mice, consistent with our observation that absolute numbers of GP66-specific GC Tfh cells have recovered in infected mice at this stage (*Figure 4G*). This timepoint corresponds roughly to the time that *P. chabaudi* parasitemia is cleared to sub-patent levels, although a low-grade chronic infection persists for several months in this model (*Stephens et al., 2012*). These data show that established *Plasmodium* infection significantly decreases expansion of B cells, establishment of germinal centers, and production of antibodies specific for a heterologous protein antigen.

**Figure 5.** Curtailed B cell expansion and antibody production in infected mice following heterologous immunization. Mice were immunized with or without concurrent *P. chabaudi* infection, as in *Figure 4*. (**A**) Representative gating and (**B**) quantification of GP-specific splenic B cells 8 days post-immunization. In A, plots were pre-gated on B cells. (**C, D**) Gating (left panels) and quantification (right) of CD138+ plasmablasts (**C**) or CD38- GL7+ GC B cells (**D**) 8 days after immunization with GP. Plots in D were pre-gated on CD138- cells. (**E**) GP-specific IgG antibody was measured by ELISA in the serum of uninfected or infected mice without immunization or 7 days after immunization with GP. (**F**) GP-specific serum antibodies (expressed as Area Under the Curve, AUC) measured by ELISA at the indicated times post-immunization. A-D represent data from three independent experiments (n=7 GP, 6 Infected + GP); E and F are pooled from two experiments. Red numbers within flow plot gates represent the frequency of cells within the gate. *, p< 0.05, **, p < 0.01, ***, p < 0.001, and ****, p < 0.0001 by one-way ANOVA with Tukey's post-test (**B, F**) or Mann-Whitney (others). n.s., not significant.

The online version of this article includes the following source data for figure 5:

**Source data 1.**

## Discussion

Our results paint a nuanced picture of DC function during *Plasmodium* infection. DCs do not take up iRBCs efficiently in vivo, compared to their uptake of soluble antigen. This relatively poor ability to capture RBC-associated antigen may contribute to the importance of B cells as APCs in this infection setting, since the majority of iRBCs were found to associate with B cells, along with RPMs. However, differential uptake is likely not the only factor biasing antigen presentation toward B cells, since DCs do acquire sufficient antigen in infected mice to activate CD4[+] T cells ex vivo (*Pouniotis et al., 2004*; *Sponaas et al., 2012*; *Sponaas et al., 2006*; *Voisine et al., 2010*; *Perry et al., 2004*). We speculate that some of this antigen may be acquired by DCs in soluble form after being shed from iRBCs (*Beeson et al., 2016*).

Although they did not capture iRBCs efficiently, DCs in both uninfected and infected mice readily took up soluble antigen and expressed the Class II and costimulatory molecules necessary to activate CD4[+] T cells. Further, although the composition of the DC subset changed during infection, the magnitude of the T cell response to soluble protein immunization was similar in both uninfected and infected mice, indicating that DCs were competent to activate CD4[+] T cells in a *Plasmodium*-infected spleen. The use of different markers to define DC subsets complicates direct comparisons between our findings and previous studies, but Sponaas et al. identified CD8[-] CD11b[+] DCs from infected mice as the most robust activators of CD4[+] T cells ex vivo *Sponaas et al., 2006*; we believe that at the timepoints examined, this subset largely overlaps with the CD11b[+] CD64[+] moDC population that we identified here as the main DC subset capturing soluble antigen in infected mice. In other contexts, moDCs have been found fully capable of antigen presentation and T cell activation (*Plantinga et al., 2013*; *Gieseler et al., 1998*).

Altogether, our findings are consistent with previous studies showing that DCs isolated from infected mice are capable of activating CD4[+] T cells in vitro. In addition, though, our work provides a readout of interactions between DCs and CD4[+] T cells within malarial hosts rather than ex vivo. This is important because the splenic architecture undergoes profound alteration in mice and humans with malaria, with dissolution of marginal zones and blurring of borders between the T cell zones and B cell follicles; it has been suggested that these architectural disruptions contribute to the observed delay in establishment of germinal centers, and subsequent failure to make optimal antibody responses (*Achtman et al., 2003*; *Stevenson and Kraal, 1989*; *Beattie et al., 2006*; *Urban et al., 2005*; *Cadman et al., 2008*; *Keitany et al., 2016*). These tissue changes have already begun to occur 5 days after infection, when we immunized mice with GP, and reach their peak during the window in which the GP-specific response is being initiated (*Achtman et al., 2003*; *Stevenson and Kraal, 1989*; *Cadman et al., 2008*; *Ryg-Cornejo et al., 2016a*). Our data suggest that even within this dramatically altered tissue environment, DCs are able to interact with both antigen and CD4[+] T cells in a way that permits robust expansion of the latter, ruling out the existence of global DC dysfunction in the malaria-inflamed spleen.

At the same time, we did observe a near-complete defect in differentiation of GP-specific Tfh and GC Tfh cells in mice that were immunized with GP 5 days after *Plasmodium* infection. We and others previously showed that ICOS-mediated interactions between B and T cells support maintenance of Tfh cells, while cognate B-T interactions are necessary for the GC Tfh subset (*Pepper et al., 2011*). In light of this, we infer that the loss of GP-specific Tfh and GC Tfh cells in *Plasmodium*-infected mice is due to disruption of both cognate and non-cognate interactions between GP-specific T cells and B cells. Here, alteration of the splenic architecture may play an important role, with the blurring of the T-B border potentially making it more difficult for activated T cells to locate B cells. The inflammatory milieu of the infected spleen may also negatively impact Tfh development; we and others have found that blocking inflammatory cytokines enhances the differentiation of Tfh cells and GC B cells during infections with other strains of *Plasmodium* (*Keitany et al., 2016*; *Ryg-Cornejo et al., 2016a*). Finally, it is interesting to speculate whether the BCR-independent binding of iRBCs that we observed may dilute or misdirect the antigen-specific B cell response in hosts with malaria. Indeed, *Plasmodium* infection is known to induce polyclonal, nonspecific B cell activation as well as activation of B cells autoreactive for RBC antigens (*Rivera-Correa et al., 2020*; *Rivera-Correa et al., 2019*; *Fernandez-Arias et al., 2016*; *Sardinha et al., 2002*). This could also contribute to the failure of GP-specific B cells to expand and support the differentiation of cognate Tfh cells.

An important question is whether the disruption to humoral responses that we observe in *Plasmodium*-infected spleens is a malaria-specific phenomenon. Daugan et al. previously found that

mice infected with LCMV simultaneous with or 4 days before heterologous immunization developed fewer immunogen-specific antibody-secreting cells and lower specific antibody titers, suggesting that other pathogens can induce similar immune dysfunction (*Daugan et al., 2016*). However, the same study found intact responses to heterologous immunization in mice infected with vesicular stomatitis virus, indicating that not all infections disrupt heterologous immunity. A separate study found that mice infected with *Plasmodium* two days *after* heterologous immunization exhibited defective antibody responses to the heterologous antigen, but immunized mice infected with LCMV or *Listeria monocytogenes* did not (*Banga et al., 2015*). Taken together with the contrasting results from *Daugan et al., 2016*, which differed in the timing of immunization relative to infection, these data may indicate that LCMV disrupts humoral responses within a narrower window of time than *Plasmodium*. Nevertheless, we favor the hypothesis that multiple pathogenic infections can interfere with the robust development of heterologous responses. The precise mechanisms of disruption appear to differ in different infections, but likely involve host-derived cytokines and chemokines, with type 1 interferons, lymphotoxin, and interferon gamma being implicated in studies using LCMV, *Toxoplasma gondii*, and *Plasmodium* respectively (*Keitany et al., 2016*; *Daugan et al., 2016*; *Glatman Zaretsky et al., 2012*). Typically in these studies, defects in the humoral response correlate with elevated inflammation and disruption of the splenic architecture; as inflammation resolves, the infection is cleared, and the tissue reorganizes, germinal centers and B cell responses also recover.

The initial motivation for this study was to investigate mechanisms underlying our previous discovery that B cells, rather than DCs, served as the dominant APC population shaping the CD4+ T cell response during *Plasmodium* infection. By expanding previously established assays of iRBC uptake to include examination of B cells, we found that the majority of iRBCs associate with splenic B cells rather than with DCs, and that B cells preferentially bind iRBCs over nRBCs. This finding is consistent with a previous study showing that B cells were the primary APCs that activated CD4+ T cell responses when antigen was delivered in nanoparticle form, whereas DCs were key for initiating responses to the same antigen in soluble form (*Hong et al., 2018*). In that study, B cell uptake of nanoparticle-associated antigen was BCR-dependent and performed by antigen-specific B cells. Here, we found that B cells can preferentially bind iRBCs independent of the BCR, although this does not preclude additional BCR-dependent binding by B cells specific for parasite antigens, such as those we have characterized extensively in previous studies (*Krishnamurty et al., 2016*; *Kim et al., 2019*; *Keitany et al., 2016*; *Thouvenel et al., 2021*). This selective but BCR-independent binding is intriguing; it suggests the existence of antigen-nonspecific interactions between iRBCs and B cells that might explain the reported activation of nonspecific polyclonal or autoreactive B cells during malaria. The literature suggests a number of possible such interaction pathways: receptors expressed by B cells might bind opsonizing antibody, complement, exposed phosphatidylserines or parasite proteins on the iRBC surface (*Vijay et al., 2021*; *Boyle et al., 2015*; *Donati et al., 2004*). Once taken up, iRBCs contain immunostimulatory ligands, such as parasite RNA and DNA, that could activate B cells through Toll-like receptor signaling, alone or in conjunction with signaling from any of the above pathways (*Baccarella et al., 2013*; *Sharma et al., 2011*; *Wu et al., 2010*; *Rivera-Correa et al., 2017*). The iRBCs used in this study were isolated from mice 6–7 days post-infection, a time when some anti-parasite antibody may be present on the iRBC surface; however, we did not observe any defect in B cell binding of iRBCs in vitro when we isolated iRBCs from µMT mice (which lack antibodies) or added an Fc receptor-blocking antibody (data not shown). Further dissection of interactions between iRBCs and B cells therefore remains an important avenue for future research.

One key aspect of our study with relevance for human health is its insight into the detrimental effects of ongoing *Plasmodium* infection on antibody responses to immunization. Such effects have been appreciated for decades, most notably through measurement of poor responses to childhood vaccinations in children from endemic regions (*Greenwood et al., 1972*; *Williamson and Greenwood, 1978*; *Banga et al., 2015*). Malaria also may affect antibody avidity (*Cadman et al., 2008*). But the mechanisms underlying these malaria-associated decreases in antibody titers have mostly remained unknown, although one paper did identify increased turnover of serum antibodies as a contributing factor (*Sardinha et al., 2002*). We recently found that blood-stage-induced inflammation disrupts the development of sporozoite-specific B cells and antibodies in mice (*Keitany et al., 2016*), perhaps providing an explanation for the relative scarcity of CSP-specific memory B cells in naturally exposed humans (*Wipasa et al., 2010*; *Jahnmatz et al., 2022*). In the present study, we have further

elucidated the cellular basis for defective antibody production during malaria: we showed that immunization with adjuvanted protein during acute *Plasmodium* infection activates immunogen-specific CD4$^+$ T cells, but fails to elicit the Tfh cells necessary to support a robust immunogen-specific B cell response, correlating with diminished B cell expansion and decreased serum antibody titers. The detrimental effect on antibody production is transient, suggesting that interactions between B and T cells eventually recover as the infection is cleared to sub-patent levels. However, in endemic areas, many clinically immune humans carry patent levels of parasites in their blood during much of the year (*Ryg-Cornejo et al., 2016b*). It remains to be seen whether asymptomatic parasitemia affects vaccine efficacy similarly to acute malaria. Even so, a strategy in which vaccine is administered together with anti-*Plasmodium* chemoprophylaxis has recently shown great promise (*Chandramohan et al., 2021*), raising our hope that a greater understanding of malaria's effects on the immune system will enable continued improvement of human protection from this and other diseases.

# Methods

**Key resources table**

| Reagent type (species) or resource | Designation | Source or reference | Identifiers | Additional information |
|---|---|---|---|---|
| antibody | Anti-B220 (RA3-6B2), rat monoclonal | BD | Cat# 563892; RRID:AB_2738470 | 1:100 dilution |
| antibody | Anti-CD138 (281-2), rat monoclonal | Biolegend | Cat# 142516; RRID:AB_2562337 | 1:100 dilution |
| antibody | Anti-CD38 (90), rat monoclonal | Biolegend | Cat# 102717; RRID:AB_2072892 | 1:100 dilution |
| antibody | Anti-GL7 (GL-7), rat monoclonal | eBioscience | Cat# 48-5902-80; RRID:AB_10854881 | 1:100 dilution |
| antibody | Anti-IgM (RMM-1), rat monoclonal | Biolegend | Cat# 406512; RRID:AB_2075943 | 1:100 dilution |
| antibody | Anti-CD4 (GK1.5), rat monoclonal | eBioscience | Cat# 19-0041-82; RRID:AB_469533 | 1:100 dilution |
| antibody | Anti-CD8 (53–6.7), rat monoclonal | BD | Cat# 553034; RRID:AB_394572 | 1:100 dilution |
| antibody | Anti-CD11b (M1/70), rat monoclonal | eBioscience | Cat# 14-0112-81; RRID:AB_467107 | 1:100 dilution |
| antibody | Anti-CD86 (GL-1), rat monoclonal | Biolegend | Cat# 105013; RRID:AB_439782 | 1:100 dilution |
| antibody | Anti-CD45.1 (A20), mouse monoclonal | BD | Cat# 560578; RRID:AB_1727488 | 1:100 dilution |
| antibody | Anti-CD45.2 (104), mouse monoclonal | BD | Cat# 560696; RRID:AB_1727494 | 1:100 dilution |
| antibody | Anti-CD11c (HL3), Armenian hamster monoclonal | BD | Cat# 553802; RRID:AB_395061 | 1:100 dilution |
| antibody | Anti-CD16/CD32 (2.4G2), rat monoclonal | BD | Cat# 553142 RRID:AB_394657 | 1:1000 dilution |
| antibody | Anti-CD64 (X54-5/7.1), mouse monoclonal | Biolegend | Cat# 139309 RRID:AB_2562694 | 1:100 dilution |
| antibody | Anti-Thy1.2 (53–2.10, rat monoclonal) | BD | Cat# 565257 RRID:AB_2739136 | 1:100 dilution |
| antibody | Anti-MHCII (M5/114.15.2), rat monoclonal | eBioscience | Cat# 56-5321-82 RRID:AB_494009 | 1:100 dilution |
| antibody | Anti-F4/80 (BM8), rat monoclonal | Biolegend | Cat# 123115 RRID:AB_893493 | 1:100 dilution |
| antibody | Anti-Siglec H (551), rat monoclonal | Biolegend | Cat# 129611 RRID:AB_10643574 | 1:100 dilution |

*Continued on next page*

*Continued*

| Reagent type (species) or resource | Designation | Source or reference | Identifiers | Additional information |
|---|---|---|---|---|
| antibody | Anti-CD3e (500A2), Syrian hamster monoclonal | BD | Cat# 553240 RRID:AB_394729 | 1:100 dilution |
| antibody | Anti-CD80 (16–10 A1), Armenian hamster monoclonal | eBioscience | Cat# 12-0801-82 RRID:AB_465752 | 1:100 dilution |
| antibody | Anti-CD19 (1D3), rat monoclonal | eBioscience | Cat# 47-0193-82 RRID:AB_10853189 | 1:100 dilution |
| antibody | Anti-CD44 (IM7), rat monoclonal | BD | Cat# 560567 RRID:AB_1727480 | 1:100 dilution |
| antibody | Anti-CD62L (MEL-14), rat monoclonal | Biolegend | Cat# 104440 RRID:AB_2629685 | 1:100 dilution |
| antibody | Anti-PD-1 (J43), Armenian hamster monoclonal | eBioscience | Cat# 11-9985-85 RRID:AB_465473 | 1:100 dilution |
| antibody | Anti-CXCR5 (2G8), rat monoclonal | BD | Cat# 560617 RRID:AB_1727521 | 1:20 dilution |
| antibody | Anti-CD71 (R17217), rat monoclonal | Invitrogen | Cat# 12-0711-83 RRID:AB_465741 | 1:200 dilution |
| antibody | Anti-CD45 (30-F11), rat monoclonal | BD | Cat# 559864 RRID:AB_398672 | 1:200 dilution |
| antibody | Anti-Ter119 (Ter119), rat monoclonal | Invitrogen | Cat# MA5-17822 RRID:AB_2539206 | 1:200 dilution |
| Chemical compound, drug | Hoescht 33342 Solution | ThermoFisher | Cat# 62249 | 1:1000 dilution |
| Peptide, recombinant protein | LCMV Glycoprotein | E.Ollman Saphire | N/A | |
| Peptide, recombinant protein | I-A(b) LCMV GP 66–77 Monomer | NIH Tetramer Core Facility | N/A | Used at 1 uM |
| Peptide, recombinant protein | Streptavidin-APC | Agilent | Cat# PJ27S-1 | |
| Strain, strain background (*Plasmodium yoelii*) | *Plasmodium yoelii* (17 X NL) GP66 | **Hahn et al., 2018** | N/A | |
| Strain, strain background (*Plasmodium chabaudi*) | *Plasmodium chabaudi* AS | MR4 Stock Center | #MRA-743 | |
| Strain, strain background (*Mus musculus*) | C56Bl/6 J (B6) mice | Jackson Labs | Stock #000664; RRID:IMSR_JAX:000664 | |
| Strain, strain background (*Mus musculus*) | CD45.1+mice | Jackson Labs | Stock #002014; RRID:IMSR_JAX:002014 | |
| Strain, strain background (*Mus musculus*) | C57BL/6-Tg(IghelMD4)4Ccg/J (MD4) mice | Jackson Labs | Stock #002595; RRID:IMSR_JAX:002595 | |
| Software, algorithm | Prism 9 | GraphPad | https://www.graphpad.com/; RRID:SCR_002798 | |
| Software, algorithm | FlowJo 10 | TreeStar | RRID:SCR_008520 | |

## Mice

Wild-type C57Bl/6 J (B6), CD45.1[+], and MD4 mice were purchased from Jackson Laboratories. B6 and CD45.1[+] mice were crossed in-house to generate CD45.1[+] CD45.2[+] mice. All mice were group-housed at the University of Washington on a twelve-hour light-dark cycle under specific pathogen-free conditions. 8–12 week old male and female mice were used for all experiments. Mice were age- and sex-matched within each experiment, and no significant differences were observed between sexes. Mice were randomly assigned to experimental groups. Sample sizes were determined by past experience. No mice were excluded from analysis. All mouse experiments were conducted with the approval of the UW Institutional Animal Care and Use Committee (Protocols

4412–01 to MFF and 4283–01 to MP) in accordance with the guidelines of the NIH Office of Laboratory Animal Welfare.

### *Plasmodium* parasites

Wild-type *Plasmodium chabaudi* AS was obtained from the MR4 Stock Center and passaged in B6 mice. Transgenic *P. yoelii* 17XNL-GP66 has been described (*Hahn et al., 2018*). For *P. chabaudi* infections, mice were injected intraperitoneally (i.p.) with $10^6$ infected RBCs. Parasitemia was monitored by thin blood smear stained with Giemsa (*Huang et al., 2015*). For immunization with irradiated parasites, *P. yoelii* GP66-infected RBCs were suspended in Alsevers solution and subjected to 30,000 rad on ice. $10^7$ irradiated iRBC were then injected i.p. into mice.

### Enrichment and labeling of iRBCs

Blood was collected by cardiac puncture from infected mice 6–7 days post-infection, when parasitemia was ~20–40%, and immediately diluted into 10 mL PBS. It was then spun, resuspended in PBS, and loaded onto pre-washed LD columns (Miltenyi). The bound fraction was washed with 4–5 mL PBS, eluted, spun down and resuspended at $5*10^7$ RBC/mL in PBS. CFSE or CellTrace Violet (Invitrogen) was added to a final concentration of 5 μM and cells were labeled for 20 min at room temperature. Labeled RBCs were washed in five volumes of cold DMEM + 10% FBS, spun, and resuspended in PBS at $10^9$ cells/mL for injection. Degree of enrichment was measured both by thin blood smear and by flow cytometry using Hoescht nucleic acid dye to identify iRBCs. To generate naïve RBCs, blood from a naïve mouse was collected and passed through an LD column. The flowthrough was collected and labeled as described for iRBCs. Injections of $10^8$ labeled RBCs were performed via retroorbital vein. In some cases, labeled iRBCs were mixed with 100 μg PE (Agilent) prior to injection.

### Detection of RBC binding during processing of spleens

CD45.2$^+$ (B6) mice were injected with labeled, enriched iRBCs as described above. After 30 min, each spleen was removed and placed into 5 mL PBS on ice along with a freshly excised spleen from an untreated CD45.1$^+$ CD45.2$^+$ mouse. After harvests were complete, each pair of spleens was manually disrupted together, filtered through nylon mesh, and placed back on ice. Flow cytometric analysis was performed as described below.

### In vitro RBC binding

Spleens were aseptically removed from naïve B6 and MD4 mice after euthanasia according to approved methods. Single-cell suspensions were made, and $2*10^6$ B6 or MD4 splenocytes were mixed with $10^7$ labeled, enriched iRBCs or nRBCs in DMEM containing 10% FBS. The combined cells were incubated for 20 min at 37 degrees, followed by ACK lysis of RBCs and flow cytometric analysis (see below).

### Protein immunizations

10 μg recombinant LCMV glycoprotein in PBS was diluted 1:1 in Sigma Adjuvant System (Sigma) and administered i.p. B and T cell responses were analyzed in spleens 8 or more days after immunization, as indicated in figure legends.

### Flow cytometry

For experiments involving analysis of DCs and macrophages, excised spleens were cut into 6–8 pieces and digested for 15 min at 37 degrees in digest buffer (RPMI + 2% FBS +10 mM HEPES + 1 mg/mL Collagenase IV + 20 μg/mL DNase I). The tissue was then manually disrupted and passed through nylon mesh to obtain single-cell suspensions. The digest step was omitted when only lymphocytes were analyzed. RBCs were lysed in ACK buffer and samples were blocked with anti-CD16/CD32 prior to labeling with antibodies. To analyze antigen-specific T cells, splenocytes were labeled with I-A(b) LCMV GP 66–77 tetramer (NIH Tetramer Core Facility) conjugated to APC (*Arroyo and Pepper, 2020*). To analyze GP-specific B cells, splenocytes were incubated first with a decoy reagent labeled in-house with APC-Dylight755 (*Taylor et al., 2012*), then with a fluorescently labeled GP tetramer (*Kim et al., 2019*). Magnetic anti-APC beads (Miltenyi) were used to enrich tetramer-positive cells after labeling (*Krishnamurty et al., 2016*; *Taylor et al., 2012*). Following enrichment, cells were incubated with

antibodies and washed. Data were collected on an LSRII or Symphony (BD) and analyzed using FlowJo software (Treestar). Antibodies are listed in the Key Resources Table.

## ELISA

Blood was collected from the submental veins of mice and serum was snap-frozen until analysis. High-binding 96 well plates (Costar) were coated with recombinant GP (5 µg/mL in PBS). After washing, serially diluted serum was applied to plates and incubated at room temperature for 2 hr or overnight at 4 °C. Plates were washed and biotinylated anti-mouse IgG (1:1000; Biolegend) was added for 1 hr at room temperature, followed by streptavidin-HRP (1:500; Jackson) for 30 min at room temperature. After a final wash, plates were developed with TMB (Invitrogen), the reaction was stopped with 1 M HCl, and absorbance was measured at 450 nm. Area under the curve (AUC) was calculated with Graphpad Prism 9 software.

## Statistical analysis

All statistics were calculated using Prism 9 software. The exact $n$ and statistical test used for each analysis are listed in each figure legend. Exact $p$-values are noted in each figure, except where $p<0.0001$, which is indicated with ****.

## Acknowledgements

We thank Kathryn Hastie for help with GP production and members of the Pepper lab for helpful discussions and assistance with sample collection.

## Additional information

### Funding

| Funder | Grant reference number | Author |
| --- | --- | --- |
| National Institutes of Health | 1R0AI118803-01A | Marion Pepper |
| National Institutes of Health | 122353SUB / 5UO1AI42001-02 | Marion Pepper |
| Burroughs Wellcome Fund | 1016766 | Marion Pepper |

The funders had no role in study design, data collection and interpretation, or the decision to submit the work for publication.

### Author contributions

Mary F Fontana, Conceptualization, Formal analysis, Investigation, Visualization, Methodology, Writing – original draft, Project administration, Writing – review and editing; Erica Ollmann Saphire, Resources; Marion Pepper, Conceptualization, Resources, Supervision, Funding acquisition, Project administration, Writing – review and editing

### Author ORCIDs

Mary F Fontana http://orcid.org/0000-0003-3630-181X
Marion Pepper http://orcid.org/0000-0001-7278-0147

### Ethics

All mouse experiments were conducted with the approval of the UW Institutional Animal Care and Use Committee (Protocols 4412-01 to MFF and 4283-01 to MP) in accordance with the guidelines of the NIH Office of Laboratory Animal Welfare.

### Decision letter and Author response

Decision letter https://doi.org/10.7554/eLife.83330.sa1
Author response https://doi.org/10.7554/eLife.83330.sa2

## Additional files

### Supplementary files
• MDAR checklist

### Data availability
All data generated or analyzed during this study are included in the manuscript, figures, and provided source data files for Figures 1–5.

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
