## [Editor Report]

Malaria is still one of the world's most deadly diseases. Here, by using animal models of malaria infection and vaccination, authors have nicely showed that Dendritic cells (DCs) have a lower ability to uptake Plasmodium (the causative agent of malaria)-infected RBCs (particulate antigen), thereby being insufficient to activate T cells. This could be a potential reason for why our bodies cannot make appropriate acquired immunity upon Plasmodium infection.

---

## [Decision Letter]

**Decision letter after peer review:**

Thank you for submitting your article "Plasmodium infection disrupts the T follicular helper cell response to heterologous immunization" for consideration by *eLife*. Your article has been reviewed by 2 peer reviewers, and the evaluation has been overseen by a Reviewing Editor and Tadatsugu Taniguchi as the Senior Editor. The following individual involved in the review of your submission has agreed to reveal their identity: Michelle J Boyle (Reviewer #2).

Essential revisions:

1) In regard to reviewer #2 conceptual concerns (from 1) to 3)), please discuss these points carefully.

2) Authors should clarify whether the impairment of B-T cell interaction is due to direct BCR interaction with iRBC, or indirect response to extrinsic factors induced by malaria infection, by using MD4 mice

3) Does *P. chabaudi* infection have any effects on B cell uptake of subsequent antigens, such as soluble antigen PE or particulate antigen CFSE-labeled P. yoelii iRBC?

*Reviewer #1 (Recommendations for the authors):*

This manuscript clearly demonstrates that murine malaria infection with Plasmodium chabaudi impairs B cells' interaction with T cells, rather than DCs interaction with T cells. The authors elegantly showed that DCs were activated, capable of acquiring antigens and priming T cells during *P. chabaudi* infection. B cells are the main APC to capture particulate antigens such as infected RBC (iRBC), while DCs preferentially take up soluble antigens. This study is important to understand how ongoing infections such as malaria may negatively affect heterologous immunizations.

Overall, the experimental designs are straightforward, and the manuscript is well-written. However, there were several limitations in this study.

Specific comments:

1. The mechanism of how the prior capture of iRBC by B cells lead to the impairment of B-T interaction was not understood. It is unclear whether the impairment of B-T cell interaction is due to direct BCR interaction with iRBC, or an indirect response to extrinsic factors induced by malaria infection.

2. Would malaria infection in MD4 mouse that carries transgenic BCR that does not recognize malaria parasite impair subsequent B cell response to HEL immunization? This may clarify whether the impairment of subsequent B cell response is BCR-specific. If malaria impairs subsequent B cell response to HEL in MD4 mouse, it might suggest that other cell types and B cell-extrinsic factors might be involved in causing the impaired B cell responses, instead of malaria affecting B cells directly.

3. MD4 mice were mentioned in the Methods in vitro RBC binding, although none of the figures described the usage of MD4 mice. This experiment data might be important to show whether RBC binding to B cells is mediated through BCR.

4. Does *P. chabaudi* infection have any effects on B cell uptake of subsequent antigens, such as soluble antigen PE or particulate antigen CFSE-labeled P. yoelii iRBC?

5. Is this phenomenon specific to malaria infection? Does malaria-irrelevant particulate immunization affect T-B interaction of subsequent heterologous immunization?

6. Despite the impaired Tfh and GC 8 days after immunization following malaria infection, Figure 5F showed GP-specific IgG eventually increased to the same level as the uninfected immunized mice on day 23. Did the authors check whether these mice had a delayed Tfh and GC response that eventually increase on day 23? Are these antibody responses derived from GC, or GC-independent response?

7. Does recovery from malaria infection by antimalarial treatment rescue the B cell response to subsequent heterologous immunization?

8. Figure 1C shows more nRBC was taken up than iRBC in B cells, but Line 142 states that "B cells bound significantly more iRBC than nRBC. Is there a mistake in the figure arrangement? Why do B cells take up for naïve RBC than iRBC?

9. Figure S1 C and D are confusing. CD45.1+ CD45.2+ mouse did not receive labeled iRBC, but why iRBC was detected as much as 40% in the spleen of this naïve mouse?

*Reviewer #2 (Recommendations for the authors):*

The data presented support the conclusions of the paper, and my concerns are largely conceptional in how we understand this data in the context of malaria infection in vaccination in endemic areas

1) The data is presented based on the idea that antigen uptake and presentation differ between particle and soluble antigens, and that during malaria infection particle uptake is more important due to circulating iRBCs. However, during parasite invasion of RBCs, the parasite sheds large amounts of antigen into the circulation, at least some of which would then be found in a soluble form in the circulation. Can the authors comment on this aspect of infection and if/how this may impact the interpretation of results? Do authors assume that any soluble antigen taken up and presented (via DCs?) during infection would be impacted as for GP66 soluble antigen? Or could an interaction on immune responses where the antigen is presented via both particle and soluble pathways?

2) Impact of particle antigen opsonisation on antigen uptake and presentation. The authors use parasites isolated from mice who have been infected for 6-7 days to investigate the ability of different subsets to update particle antigens. At this time point, have mice developed antibody responses that opsonise these parasites, or are antibody levels low and parasites opsonised? Would opsonised parasites, such as those coated with sera from children in a setting of chronic infection, have a different pattern/ability to be opsonised by different immune cell subsets? And/or would opsonisation change how the DC and other cell types are processing/presenting antigens? While these issues could be addressed experimentally either now or in the future, the manuscript should at least consider this issue because, during a human infection in areas of high exposure, individuals are likely to have reasonable levels of antibodies with opsonised parasites circulating.

3) While authors show that malaria infection disrupts the response to soluble antigens, the relevance directly to vaccination should be considered carefully, specifically because vaccines of soluble antigens are largely given alongside adjuvants which also will modulate DC function. Again, this could be addressed experimentally now or in the future, but definitely should be mentioned and considered when interpreting the results.

Prior to publication, I suggest the following edits/queries for clarity.

Is the legend in Figure 1 correct? Line 142 states "B cells bound significantly more iRBCs than nRBCs …" but the figure is the opposite as labelled. I assumed that the labels have been switched, but best to check and match back to the text as needed.

In Figure 1 – suggestion to also analyse the data as RBC+ (% of total cell subset) – ie RBC+ DCs/% of total DCs, to understand the relative capacity of each subset to uptake antigen as a % of all those cells in that subset.

In Figure 2 – it looks like that are hardly any double positive cells – ie those which have taken up both PE and iRBCs at the same time – is this expected and consistent or just in the gating example? Does it suggest the specialisation of both B cells and DCs to be better able to uptake soluble or particle antigens in some way or suggestive of different subsets?

I don't understand Figure 3C – why is the PE+ DCs in the naive mice 0? Additionally, the text states in line 226 "we observed equivalent or slightly higher frequencies" but Figure 3C has a clear significantly increased uptake in infected mice.

At day 23, when antibody levels to soluble antigen (Figure 5F) have recovered to levels of uninfected mice – are robust Tfh/GC B cells detected? Ie, is there evidence of germinal centre recovery? Is this due to long-term soluble antigen in DCs that is able to then activate naïve T/B cells down the track?

Figure 5 – shows reduced IgG, but would be beneficial to also look at the specific subclasses of these antibodies, particularly given the importance of cytophilic subclasses in protection.

---

## [Author Response]

Essential revisions:1) In regard to reviewer #2 conceptual concerns (from 1) to 3)), please discuss these points carefully.

We have addressed these points with additional discussion and references to the relevant literature; please see our responses to Reviewer #2, Public Review Points 1-3, below.

2) Authors should clarify whether the impairment of B-T cell interaction is due to direct BCR interaction with iRBC, or indirect response to extrinsic factors induced by malaria infection, by using MD4 mice

We have addressed this point extensively in our responses to Reviewer #1, Specific Comments 1 and 2, below.

3) Does P. chabaudi infection have any effects on B cell uptake of subsequent antigens, such as soluble antigen PE or particulate antigen CFSE-labeled *P. yoelii* iRBC?

We now provide data showing the effect of *P. chabaudi* infection on B cell uptake of soluble PE; Please see our response to Reviewer #1, Specific Comment #4.

Reviewer #1 (Recommendations for the authors):This manuscript clearly demonstrates that murine malaria infection with Plasmodium chabaudi impairs B cells' interaction with T cells, rather than DCs interaction with T cells. The authors elegantly showed that DCs were activated, capable of acquiring antigens and priming T cells during P. chabaudi infection. B cells are the main APC to capture particulate antigens such as infected RBC (iRBC), while DCs preferentially take up soluble antigens. This study is important to understand how ongoing infections such as malaria may negatively affect heterologous immunizations.Overall, the experimental designs are straightforward, and the manuscript is well-written. However, there were several limitations in this study.Specific comments:1. The mechanism of how the prior capture of iRBC by B cells lead to the impairment of B-T interaction was not understood. It is unclear whether the impairment of B-T cell interaction is due to direct BCR interaction with iRBC, or an indirect response to extrinsic factors induced by malaria infection.

We believe we have carefully demonstrated that impairment of B-T interactions does not require specific BCR-antigen interactions between B cells and iRBCs (for a complete explanation of this point, please see the response to the next comment). However, the question remains whether direct, antigen-nonspecific iRBC-B cell interactions (i.e., not mediated by the BCR) or additional extrinsic factors, or a combination, are responsible for the observed defects in Tfh and GC B cell populations.

Existing studies from other infection models are informative in answering this question. Daugan et al. (Front Immunol 2016; PMID 27994594) previously published experiments similar to ours, but used LCMV instead of *Plasmodium*. That is, they immunized uninfected or LCMV-infected mice with the well-studied immunogen NPP-CGG and measured NP-specific antibody production and other parameters. They found that LCMV infection concurrent with immunization (or 4-8 days before) significantly decreased the numbers of NP-specific splenic antibody-secreting cells and IgG1 titers, and caused major disruptions to splenic architecture. These defects were shown to require type I interferon (T1IFN) signaling in B cells. However, T1IFN is unlikely to be solely responsible for the observed phenotypes, because simultaneous infection with VSV, another virus that also induces T1IFN, did not cause any defects in NP-specific antibody production. Contrasting with the work of Daugan et al., Banga et al. (PloS One 2015; PMID 25919588) found that infecting with LCMV (or with *Listeria monocytogenes*) two days after heterologous immunization did not disrupt immunogen-specific responses, whereas *P. yoelii* did. Examining both these studies, we hypothesize that both LCMV and *Plasmodium* infections can disrupt humoral responses, but that LCMV does so within a narrower time frame, thereby yielding different results depending on whether infection comes a few days before or a few days after immunization.

Complementing these studies of heterologous immunization, additional publications have reported that cytokines induced by several different pathogenic infections drive disruption of germinal centers and decreases in antibody titers specific for the pathogen itself, often correlated with disordered splenic architecture. Glatman Zaretsky et al. (Infect Immun 2012; PMID 22851754) showed that *Toxoplasma gondii* infection causes transient disruption of splenic architecture and loss of defined GCs by microscopy. These defects were partially due to decreased lymphotoxin expression by B cells, and were rescued by a lymphotoxin receptor agonist. Similarly, we previously reported that blood-stage *Plasmodium* infection disrupted germinal center responses to a *Plasmodium* liver-stage antigen (Keitany et al. Cell Rep 2016; PMID 28009289). In this context, however, the same lymphotoxin receptor agonist had no effect on GCs; instead, blockade of the pro-inflammatory cytokine interferon γ partially restored antibody responses to the liver-stage antigen. Overall, we favor the hypothesis that several different pathogens can disrupt GCs and antibody responses indirectly by inducing inflammation and a disordered splenic environment; however, the precise mechanisms of disruption likely differ from infection to infection, with different cytokines or other effectors playing key roles in some but not other settings. Importantly, not all pathogens disrupt antibody production, since again, infection with VSV or *L. monocytogenes* did not affect immunogen-specific titers in immunized mice (Daugan Front Immunol 2016; Banga et al. 2015). We have now addressed this topic at length in the Discussion (lines 399-418).

The existence of indirect, inflammation- or cytokine-related mechanisms that may interfere with germinal center formation and antibody production does not preclude additional direct interactions between B cells and iRBCs that might also affect B cell function. We address this possibility more fully in the response to the next comment.

2. Would malaria infection in MD4 mouse that carries transgenic BCR that does not recognize malaria parasite impair subsequent B cell response to HEL immunization? This may clarify whether the impairment of subsequent B cell response is BCR-specific. If malaria impairs subsequent B cell response to HEL in MD4 mouse, it might suggest that other cell types and B cell-extrinsic factors might be involved in causing the impaired B cell responses, instead of malaria affecting B cells directly.

The question of whether the impairments we observe require BCR-specific interactions with iRBCs is an important one. However, we believe that the experiment the reviewer proposes to address this question has technical limitations; further, we assert that we have already provided data to address a requirement for BCR specificity.

With regard to the proposed experiment of immunizing MD4 mice with HEL in the presence or absence of malaria infection: MD4 mice, in which B cells express a transgenic receptor specific for HEL, can be expected to mount a massive, monoclonal response to direct immunization with HEL that would be very different from the physiological context of a polyclonal B cell population. We are doubtful that this experimental setup would be informative for the question at hand, especially because we are studying the effects of B-Tfh interactions, which are already limiting in the physiological setting of a polyclonal B cell response, but would be massively unbalanced in an MD4 mouse where all B cells express the receptor for HEL.

Usually, investigators studying MD4 B cell responses generate a more physiological setting by adoptively transferring a small but detectable number of MD4 transgenic B cells into a mouse with a normal polyclonal B cell population, and immunizing that mouse. We maintain that this approach is essentially what we have done in our study, except that instead of using transferred. transgenic cells to identify a B cell population of known specificity, we have used tetramers to detect a specific population of endogenous B cells in a polyclonal setting. By examining GP-specific B cells in our immunization experiments, we restricted our analysis to B cells that could not have had any BCR-mediated, antigen-specific interactions with iRBCs (because the GP antigen is not present in the iRBCs; it is delivered as a soluble protein antigen, 5 days after initiation of infection). Because we see dysfunction in the GP-specific T and B cell populations despite the absence of this antigen within iRBCs, we can conclude that the disruptions to these populations are not due to antigen-specific iRBC-BCR interactions.

We do also show (using MD4 B cells in Figure S1B) that selective interactions between iRBCs and B cells do not require an antigen-specific BCR. Thus, it is still possible that direct interactions between iRBCs and B cells (that are independent of antigen binding to the BCR) are responsible for disrupting subsequent adaptive responses, perhaps in addition to the more indirect factors that we discuss in the response to Comment #1 above. We are very interested in this possibility, which is discussed in lines 428-436 of the manuscript. But the use of MD4 B cells would not address this specific question. Instead, we would need to identify an alternative pathway or receptor that mediates the iRBC-B cell interaction, and study the effects of blocking that pathway on downstream adaptive responses. We have spent considerable time and energy on this question, but have not yet been able to identify such a pathway; this remains a matter for further study.

3. MD4 mice were mentioned in the Methods in vitro RBC binding, although none of the figures described the usage of MD4 mice. This experiment data might be important to show whether RBC binding to B cells is mediated through BCR.

Cells from MD4 mice were used in Figure S1B to show that in vitro binding of iRBCs to B cells did not require interaction with an antigen-specific BCR. We agree that this is an important point and have revised the text (lines 152-156) to outline it more clearly.

4. Does *P. chabaudi* infection have any effects on B cell uptake of subsequent antigens, such as soluble antigen PE or particulate antigen CFSE-labeled *P. yoelii* iRBC?

We examined uptake of PE by B cells in *P. chabaudi*-infected mice (5 days post-infection) compared to naïve mice. There was a trend towards increased uptake in the infected mice, but this difference was not significant. These data are taken from the same samples that did reveal a significant increase in PE uptake by DCs in infected mice (Figure 3C). We have now included the B cell data in the paper as Figure 3D, and discussed them in lines 231-232.

5. Is this phenomenon specific to malaria infection? Does malaria-irrelevant particulate immunization affect T-B interaction of subsequent heterologous immunization?

We do not believe this phenomenon is specific to malaria infection; please see the extensive discussion of this point in the response to Comment #1 above. We would hypothesize that malaria-irrelevant particle immunization (as with nanoparticles) would not affect T-B interactions for subsequent heterologous immunizations, however, since the disruption seems to be associated with the massive inflammation and splenic disorganization that occurs following certain infections.

6. Despite the impaired Tfh and GC 8 days after immunization following malaria infection, Figure 5F showed GP-specific IgG eventually increased to the same level as the uninfected immunized mice on day 23. Did the authors check whether these mice had a delayed Tfh and GC response that eventually increase on day 23? Are these antibody responses derived from GC, or GC-independent response?

We have now examined GP-specific T cell numbers and polarization between days 23 and 35 post-immunization. We found that although a defect persists in the percentage of GP66-specific T cells that exhibit a GC Tfh phenotype at later timepoints, the absolute number of GC Tfh cells is not significantly defective in infected mice at these times. Concurrently there is a slight (though nonsignificant) increase in the total numbers of GP66+ T cells in the infected mice; we believe that this modest overall expansion permits recovery of the GC Tfh population numbers despite the continued defect in their frequency. These findings are consistent with our observation that antibody levels recover in infected mice by 3 weeks post-infection. We have added these data to Figure 4 (E-G) and discuss them in lines 283-293.

7. Does recovery from malaria infection by antimalarial treatment rescue the B cell response to subsequent heterologous immunization?

We have shown previously that drug-mediated clearance of blood-stage *Plasmodium* infection restores GC and antibody responses to a liver-stage-specific antigen, which normally are disrupted by emergence of the blood-stage (Keitany et al. Cell Rep 2016). We have also shown that antimalarial drug treatment restores GC responses in mice lacking the innate immune sensor CGAS, which have higher parasitemia, exacerbated splenic disruption, and diminished GC responses following *P. yoelii* infection (Hahn et al., JCI Insight 2018). Based on these results we hypothesize that drug-mediated clearance of blood-stage infection would also rescue B cell responses to heterologous immunization.

8. Figure 1C shows more nRBC was taken up than iRBC in B cells, but Line 142 states that "B cells bound significantly more iRBC than nRBC. Is there a mistake in the figure arrangement? Why do B cells take up for naïve RBC than iRBC?

The symbols in the figure legend were switched in error; the filled circles are actually iRBC+ and the outlined circles are nRBC+. We regret the error and appreciate the reviewer bringing it to our attention. We have corrected the figure.

9. Figure S1 C and D are confusing. CD45.1+ CD45.2+ mouse did not receive labeled iRBC, but why iRBC was detected as much as 40% in the spleen of this naïve mouse?

The experiment depicted in Figures S1 C and D was designed to test whether B cells actually bound injected iRBCs in vivo, or whether the binding occurred during processing of the tissue. With this experimental setup (injecting labeled iRBCs into CD45.2+ mice, then excising and disrupting the spleen together with an untreated CD45.1+ CD45.2+ spleen), iRBC signal from in vivo uptake should be observed only in CD45.2+ splenocytes, whereas iRBC binding that occurs during tissue processing will be distributed between the two genotypes. Thus, the ~40% of iRBC signal observed in CD45.1+ CD45.2+ B cells leads us to conclude that much of the observed B cell binding from our in vivo experiments occurs during processing, as we state in the text (lines 151-152). Even so, in vitro experiments clearly show that B cells selectively bind iRBCs over naïve RBCs in a setting where processing is not a confounder (Figure S1B). To clear up any confusion, we have expanded the description of the experiment and its interpretation in the Supplemental Figure Legend.

Reviewer #2 (Recommendations for the authors):The data presented support the conclusions of the paper, and my concerns are largely conceptional in how we understand this data in the context of malaria infection in vaccination in endemic areas1) The data is presented based on the idea that antigen uptake and presentation differ between particle and soluble antigens, and that during malaria infection particle uptake is more important due to circulating iRBCs. However, during parasite invasion of RBCs, the parasite sheds large amounts of antigen into the circulation, at least some of which would then be found in a soluble form in the circulation. Can the authors comment on this aspect of infection and if/how this may impact the interpretation of results? Do authors assume that any soluble antigen taken up and presented (via DCs?) during infection would be impacted as for GP66 soluble antigen? Or could an interaction on immune responses where the antigen is presented via both particle and soluble pathways?

This is an important point and we have now discussed it further in the text (lines 111-115, 204-210, and 356-357). In our previously published study, where we extensively characterized CD4 T cell responses to the GP66 epitope expressed by *P. yoelii*, the epitope was fused to a parasite protein (Hep17) that localizes to the parasitophorous vacuole membrane, and so we do assume that the majority of this antigen is encountered by APCs in the context of an iRBC, rather than shed in soluble form. In contrast, some merozoite surface antigens such as cleaved MSP1 are shed copiously from the parasite coat upon RBC invasion, and therefore would be expected to exist in soluble as well as parasite-associated form.

Unfortunately, our laboratory does not currently have tetramer reagents or access to transgenic mice that would allow us to assess T cell responses specific for shed or soluble parasite antigens. But a previous study from Stephens et al. (Blood 2005; PMID 15890689) reported that T cells with a transgenic TCR specific for an epitope in the shed portion of MSP1 could boost antibody production when transferred into T cell-deficient mice infected with *P. chabaudi*, suggesting that at least some fraction of the MSP1-specific T cells differentiate into T helper cells capable of supporting B cell activity. However, antibody production was significantly delayed in this setting compared to its usual kinetics in wild-type mice. Further side-by-side characterization would be needed to assess differentiation of these MSP1-specific transgenic T cells during infection, and determine whether they are primed from B cells or from DCs (or a combination).

We will note that we *have* extensively characterized B cell responses to MSP1 during both infection and immunization. While robust and T-dependent, MSP1-specific B cell responses in infected mice are delayed relative to their kinetics in mice immunized with recombinant MSP1 or other protein antigens. This may indicate that MSP1-specific T cell activation or cognate B-T interactions are defective in infected mice relative to immunized mice, despite the presumed presence of soluble (shed) MSP1 during infection. If this is the case, it suggests that the defects we describe in the current manuscript exist for both particle-associated and soluble parasite antigens. However, as we mentioned above, a careful characterization of MSP-1-specific T cell differentiation is needed to really understand this, and that requires additional tools that we can’t easily access at this time.

2) Impact of particle antigen opsonisation on antigen uptake and presentation. The authors use parasites isolated from mice who have been infected for 6-7 days to investigate the ability of different subsets to update particle antigens. At this time point, have mice developed antibody responses that opsonise these parasites, or are antibody levels low and parasites opsonised? Would opsonised parasites, such as those coated with sera from children in a setting of chronic infection, have a different pattern/ability to be opsonised by different immune cell subsets? And/or would opsonisation change how the DC and other cell types are processing/presenting antigens? While these issues could be addressed experimentally either now or in the future, the manuscript should at least consider this issue because, during a human infection in areas of high exposure, individuals are likely to have reasonable levels of antibodies with opsonised parasites circulating.

We ourselves have been very interested in the question of whether host antibodies (or other host factors such as complement) might affect uptake of iRBCs. As the reviewer notes, the iRBCs we use in our experiments are taken from mice 6-7 days post-infection, at which time some amount of anti-parasite antibody has developed. We spent a considerable amount of time trying to measure effects of opsonizing antibody, or even deposited complement, on uptake of iRBCs. However, we did not see any change in B cell binding of iRBCs in vitro when we blocked complement receptor with anti-CD21; blocked antibody receptors (Fc receptors) with anti-CD16/CD32 or excess normal mouse serum; or used iRBCs taken from complement-depleted mice (treated with cobra venom factor) or from uMT mice (which entirely lack B cells and antibody). Thus, we have not been able to find any role for opsonizing antibody (or complement) in iRBC uptake. We have not included these experiments in the manuscript because they yielded only negative data, and we were not able to design positive controls robust enough to give us confidence that the experiments were technically sound (and therefore that the negative results were due to the underlying biology and not to experiment failure). We have added a discussion point about this issue (lines 438-442).

3) While authors show that malaria infection disrupts the response to soluble antigens, the relevance directly to vaccination should be considered carefully, specifically because vaccines of soluble antigens are largely given alongside adjuvants which also will modulate DC function. Again, this could be addressed experimentally now or in the future, but definitely should be mentioned and considered when interpreting the results.

Whenever we performed soluble protein immunizations to examine adaptive immune responses in this study, the immunogen was delivered in adjuvant, specifically Σ Adjuvant System (SAS), as described in the Methods. This adjuvant contains the Monophosphoryl Lipid A component from *Salmonella* in an oil-water emulsion, and as such, its formulation is at least roughly similar to the AS01 adjuvant used in Mosquirix (RTS,S), the only licensed anti-malaria vaccine, as well as other vaccines currently used in humans. SAS has been shown to elicit very high titers of neutralizing antibodies in mice (Sastry et al., PloS One 2017, PMID 29073183). Therefore our results should have relevance for vaccination in humans. We have modified the manuscript text (lines 454-455) to highlight that in this study, protein immunogens were administered with adjuvant.

Prior to publication, I suggest the following edits/queries for clarity.Is the legend in Figure 1 correct? Line 142 states "B cells bound significantly more iRBCs than nRBCs …" but the figure is the opposite as labelled. I assumed that the labels have been switched, but best to check and match back to the text as needed.

The symbols in the figure legend were switched in error; the filled circles are actually iRBC+ and the open circles are nRBC+. We regret the error and appreciate the reviewer bringing it to our attention. We have corrected the figure.

In Figure 1 – suggestion to also analyse the data as RBC+ (% of total cell subset) – ie RBC+ DCs/% of total DCs, to understand the relative capacity of each subset to uptake antigen as a % of all those cells in that subset.

The data are presented this way (as RBC+ % of total cell subset) in Figure 2C.

In Figure 2 – it looks like that are hardly any double positive cells – ie those which have taken up both PE and iRBCs at the same time – is this expected and consistent or just in the gating example? Does it suggest the specialisation of both B cells and DCs to be better able to uptake soluble or particle antigens in some way or suggestive of different subsets?

We consistently observed this phenomenon across all our experiments—i.e. that very few cells took up both PE and iRBCs. We can suggest at least three factors that may underlie this: (1) an intrinsic relative specialization of B cells and DCs at picking up soluble versus particulate antigen, which we describe and characterize in Figure 2 and lines 186-192; (2) spatial organization within the spleen that may affect movement of, or access to, soluble versus particulate antigen; and (3) the overall very low frequency of splenocytes that pick up either label, which statistically would also result in an even lower frequency of cells picking up both labels. These factors are not mutually exclusive. We now discuss this point explicitly (lines 172-175).

I don't understand Figure 3C – why is the PE+ DCs in the naive mice 0? Additionally, the text states in line 226 "we observed equivalent or slightly higher frequencies" but Figure 3C has a clear significantly increased uptake in infected mice.

The absolute number of DCs that picked up PE in naïve mice in these experiments was between 500-5000 cells per mouse. We had graphed these numbers on a linear scale with the y- axis tick marks representing the indicated number multiplied by 10^-4 (which we indicated in the y-axis label). In this format, the values did appear very close to zero, especially next to the data points from infected mice. For better resolution of the data, we have changed the y-axis of this graph to a log scale in the revision.

Regarding the reviewer’s point about line 226:

“we had not wanted to overstate the increase in PE uptake in infected mice, since four of the six mice in this group were only slightly higher than in the naïve group while two additional mice had much higher uptake. However, the reviewer is correct that the statistics show a significant increase in the affected mice, so we have revised our text to state this clearly (line 229).”

At day 23, when antibody levels to soluble antigen (Figure 5F) have recovered to levels of uninfected mice – are robust Tfh/GC B cells detected? Ie, is there evidence of germinal centre recovery? Is this due to long-term soluble antigen in DCs that is able to then activate naïve T/B cells down the track?

Please see our response to Reviewer 1, Specific Comment #6, with regard to recovery of Tfh cells by Day 23 and beyond. We have not examined GC B cell numbers at this later timepoint, but the recovered production of IgG in infected mice (Figure 5F) by this time, combined with the presence of near normal numbers of Tfh cells (Figure 4G), strongly suggests that germinal centers have recovered as well. Since total antigen-specific T cell numbers are intact in these mice even at early timepoints, we favor the hypothesis that resolving inflammation permits resumption of normal cognate and non-cognate interactions between B and T cells that allows recovery of germinal centers.

Figure 5 – shows reduced IgG, but would be beneficial to also look at the specific subclasses of these antibodies, particularly given the importance of cytophilic subclasses in protection.

We agree that it would be interesting to examine specific subclasses of IgG in this setting. Unfortunately, we do not have any more stored serum samples that would facilitate the rapid assessment of this question. However, we have performed extensive analyses of antigen-specific IgG+ B cells in mice with malaria and found that the great majority of the class-switched population is dependent on T cell help (Harms Pritchard and Pepper, manuscript under revision). Thus, we would hypothesize that all IgG subclasses elicited by our immunization would be affected in hosts with malaria, since our model is that B-T interactions are severely disrupted in these mice.